# Satisfying Basic Psychological Needs among People with Complex Support Needs: A Self-Determination Theory-Guided Analysis of Primary Relatives' Perspectives

Jacqueline M. van Tuyll van Serooskerken [1,*], Agnes M. Willemen [1], Anne de la Croix [2], Petri J. C. M. Embregts [3] and Carlo Schuengel [1]

[1] Department of Clinical Child and Family Studies, Faculty of Behavioural and Movement Sciences, Vrije Universiteit Amsterdam, 1081 BT Amsterdam, The Netherlands; a.m.willemen@vu.nl (A.M.W.); c.schuengel@vu.nl (C.S.)

[2] Department of Research in Education, VUmc School of Medical Sciences, Amsterdam Universitair Medische Centra, 1081 HV Amsterdam, The Netherlands; a.delacroix@amsterdamumc.nl

[3] Department of Tranzo, Tilburg School of Social and Behavioral Sciences, Tilburg University, 5037 DB Tilburg, The Netherlands; p.j.c.m.embregts@tilburguniversity.edu

\* Correspondence: j.m.van.tuyll@vu.nl; Tel.: +31-20-59-82849

**Abstract:** *Background*: The fulfilment of basic psychological needs (BPNs) is seen as an integral part of human self-determination, subjective wellbeing, and overall quality of life. However, the meaning of these psychological constructs for individuals with the most extensive support needs remains elusive. *Methods*: Primary relatives of nine people diagnosed with severe or profound intellectual and multiple disabilities were interviewed about their perceptions of autonomy, competence, and relatedness regarding their family member with complex care needs, and about the ways in which they tried to support their family member in fulfilling specific BPNs. The interview analysis followed a grounded theory with the sensitizing concepts approach. *Results*: The relatives assigned important meaning to the BPNs, providing insights into their subtle nature, their implicit drivers, and how they were experienced. The relatives also identified serious challenges in detecting, clarifying, and creating opportunities for BPNs. *Conclusions*: The themes in the relatives' perspectives can be summarized into a conceptual framework that may contribute to better mutual understanding between people with complex care needs, their relatives, and healthcare providers.

**Keywords:** people with extensive and complex support needs; primary relatives; basic psychological needs; self-determination; grounded theory; sensitizing concepts

## 1. Introduction

Supporting children in becoming individuals with a strong sense of direction who strive after desires relating to what they want in life can be challenging for caregivers, especially when communication is difficult and children require long-term support to meet their needs [1]. In particular, caregivers of children with extensive and complex support needs repeatedly struggle with questions like "what is it that my child wants?" and "how do I get my child to achieve this?", which is reflected in their desire for greater knowledge about how to support their child's needs, development, and wellbeing [2]. In the current paper, "complex support needs" refer to persons who have been described in the literature as having severe or profound intellectual and multiple disabilities (SPIMD) [3]. This combination of severe or profound cognitive and additional impairments—such as motor, sensory, communication and physical health problems—causes a permanent dependency on others in all aspects of daily life [4]. Taking care of and raising children with complex support needs thus has a huge impact on the lives of relatives, and as such potentially changes families' future prospects [5–7]. Understanding the perspectives of relatives

regarding the psychological need satisfaction and motivation (i.e., self-determination) of their family member with complex support needs may therefore not only reveal unique insights but may also increase the relevance of theories on self-determination.

The use of the self-determination construct in the disability field started in the late twentieth century and expanded exponentially in the past few decades [8,9]. This increase in attention for self-determination in the care and education of children and adults with disabilities corresponds to the first principle of the Convention on the Rights of Persons with Disabilities, which defines universal respect for one's own autonomy [10]. Wehmeyer et al. [8] argued that an action can be seen as self-determining when it is taken by a person who acts volitionally on the basis of his or her own desires. In addition, self-determination is not an inborn skill but rather develops and is maintained in interaction between the person and the environment [11]. Two major explanatory frameworks for the emergence of self-determination are Self-Determination Theory (SDT) and Causal Agency Theory (CAT). SDT is a widely known meta-theory on the interaction between social and environmental contexts, human action motivation, and psychological needs satisfaction [12]. CAT focuses on the causal action sequence of how people eventually become self-determined [13]. According to SDT, self-regulation and intrinsic motivation follow from opportunities to fulfill three innate basic psychological needs (BPNs): autonomy, competence, and relatedness [14,15]. In brief, the *need for autonomy* refers to people's desire to experience self-endorsed regulation [16], the *need for competence* refers to people's desire to feel effective in what they do [17], and the *need for relatedness* refers to people's desire to seek connection with significant others [18].

The satisfaction of the three BPNs in combination with a supportive environment enhances subjective wellbeing and overall quality of life, while the thwarting of those needs accomplishes the opposite [11,14,19]. Research with people with intellectual disabilities indicates similar links between self-determination and health-related outcomes to the general population. A study by Frielink et al. [20] with adults with mild to borderline intellectual disabilities, for example, showed positive associations between perceived autonomy support, autonomous motivation, the satisfaction of the three BPNs, and psychological wellbeing. The Quality of Life model for people with intellectual disabilities by Schalock and Verdugo [21] emphasizes the importance of "self-determination" or "self-determined behavior" as one of eight indicators of health-related quality of life. However, despite its importance, Wehmeyer [9] concluded from the available empirical evidence that persons with intellectual and developmental disabilities show less self-regulatory behavior compared to their peers. Vicente et al. [22], for example, found that the level of support needs was negatively associated with self-determination, and Carter et al. [23] concluded that people with severe intellectual disabilities had minimal self-determination capacity levels.

A lower level of self-determination in people with support needs is often attributed to limitations in their cognitive abilities. However, contextual factors may be important as well, such as living in a controlling environment, being dependent on others to create opportunities for self-regulated behavior, and a lack of adequate support [24,25]. It is possible, for example, that the quality of care is negatively influenced by the presence of stigmatizing perceptions of support staff about people with high support needs [26]. Martínez-Tur et al. [27] found that the positive attitudes of relatives towards self-determination were associated with higher frequencies of self-determined behaviors of family members with intellectual disabilities. Thus, in order to understand how the social environment facilitates or hampers the self-determination of people with complex support needs, it may be helpful to clarify and describe the perspectives of those most involved regarding this elusive concept.

There are currently few studies on self-determination specifically in people with the most extensive support needs. One explanation could be that self-determination is often misinterpreted as "having independent control" over one's own life or being capable of making considered and informed decisions, when in fact it refers to "acting volitionally" [28]. Another explanation may be the lack of suitable and reliable measures of self-determination for people with low-level cognitive functioning [3]. Furthermore, most of the studies in this

population that do focus on self-determination are mainly intervention-oriented and aimed at improving specific aspects. Examples are studies on decision-making regarding everyday issues [29], independent living and leisure skills [30], metacognition and self-regulation [31], and assisted acts of self-determination using microswitch technology [32]. There is little literature on person-specific interpretations of BPN satisfaction and motivation that go along with self-determination.

The limited scientific understanding of the perceived meaning of constructs within SDT for people with the most extensive support needs may directly hamper effective professional support. It may also hamper support indirectly, as it complicates working with others (e.g., parents) who advocate on behalf of the person with the complex care needs [33]. Studying the perspectives of relatives who act as a sounding board for those with the most extensive support needs may therefore serve the dual purpose of providing an important indication as to what self-determination might mean while also pointing towards opportunities for support. Therefore, one study aim was to explore the meaning that primary relatives ascribe to satisfaction and motivation for the three BPNs (i.e., autonomy, competence, and relatedness) for their family member with complex support needs. The other aim was to identify what relatives see as necessary for supporting self-determination.

## 2. Materials and Methods

### 2.1. Design

The SDT claims that need fulfilment leads to autonomous motivation, laying the foundation for self-determination [14,34]. Consistent with this theory, the current qualitative study explored primary relatives' perceptions regarding the BPN satisfaction and motivation of their family member with complex support needs using semi-structured interviews. The study was designed according to a grounded theory and sensitizing concepts approach in which specific constructs or interests guide qualitative data collection and analysis [35,36]. The sensitizing concepts were the basic psychological needs of "autonomy", "competence", and "relatedness" [14].

### 2.2. Data Sources and Participants

The participants were primary relatives of people diagnosed with severe or profound intellectual and multiple disabilities. Parents were recruited because of their unique, experiential, and crucial knowledge about the care and support of their child with special needs [33]. Siblings were recruited because they also play a significant role in the life of their family member with extensive support needs, especially when parents become older, have disabilities themselves, or pass away [37]. In order to be included, the participants had to have a family member at least 3 years old, diagnosed with severe or profound intellectual disability (i.e., IQ score < 35–40 points or a developmental age < 5 years) in combination with additional disabilities such as motor, sensory, communication, physical health problems. They themselves had to fulfil an active role in the life of this family member (e.g., as their parent or legal representative), speak Dutch, and be at least 18 years old. Family members with complex support needs were included whether they lived with participants or in a care facility.

The participants were recruited through several Dutch care and client advocacy organizations that support people with disabilities and their caregivers. These organizations shared information about the study in their newsletter, on their website, and through their support staff. In addition, conventional and social media were used. When interested, the relatives left their contact information on the study website, after which the researchers contacted them and fully explained the study, consent form, and measures taken to ensure confidentiality. In order to check if the participants met the inclusion criteria, the participants were asked about complex support needs in divergent domains, indicating dependency on others for all aspects of physical care, health, and safety [3]. The participants also completed a paper-and-pencil survey prior to the interview on the demographic and additional characteristics of themselves and their family member with

extensive support needs (e.g., aetiology, adaptive, communication, socialization, daily living, and motor behaviors). In case of doubt, the participants were contacted for more information. Primary relatives of nine people with extensive support needs were willing to participate, available, and met all of the inclusion criteria. After written consent was collected, practical considerations and participant availability determined the order of interviews, and the interview appointments were made. Finally, the researchers offered the participants the opportunity to have their family member present during the interview.

Interviews were held with the mother only ($n = 5$), both parents (i.e., mother and father) simultaneously ($n = 2$), and a sister ($n = 2$). All of the participants knew their family member all their lives. The educational level of the participants varied from secondary education to doctorate. The mean age of the people with complex support needs was 27.17 years ($SD = 3.49$), ranging from 7 to 63 years old. Eight participants lived with their family member in the same household. The family member of one participating sister lived in a care facility. Four family members had visual impairments that could not be corrected by glasses or contact lenses. Three had auditory impairments that could not be corrected with a hearing aid. At the time of study entry, two family members received treatment for their behavioral and psychological problems, and six received medical treatment for specific physical health problems (see Table 1 for more details).

**Table 1.** Demographic summary.

| | Participants | | | Family Members with Complex Support Needs | | | |
|---|---|---|---|---|---|---|---|
| Participant Pseudonym | Relationship | Primary Caregiver(s) of Family Member with Complex Support Needs | Primary Caregiver(s) of Others Next to Family Member with Complex Support Needs | Gender | Age Group (Years) | Other Health Issues [1] | Living Arrangement |
| R1 | Mother | √ | √ | Male | Middle childhood (6–12) | √ | Family home |
| R2 | Both parents | √ | √ | Male | Middle childhood (6–12) | √ | Family home |
| R3 | Mother | √ | × | Female | Adult (21+) | √ | Family home |
| R4 | Mother | √ | √ | Female | Adolescence (13–21) | √ | Family home |
| R5 | Mother | √ | √ | Female | Adolescence (13–21) | √ | Family home |
| R6 | Both parents | √ | √ | Male | Middle childhood (6–12) | √ | Family home |
| R7 | Sister | × | - | Male | Adult (21+) | √ | Group home |
| R8 | Sister | √ | × | Female | Adult (21+) | × | Family home |
| R9 | Mother | √ | √ | Female | Adolescence (13–21) | × | Family home |

[1] The presence of one or more severe additional health issues within the visual, auditory, behavioral/psychological, or physical health domain for which the family member was being treated at the time of study participation.

The research team consisted of five people from the fields of psychology, developmental psychopathology, child development, qualitative and quantitative research, intellectual and physical disability, policy, and nursing. One team member was also a parent of a child with complex support needs. Prior to the study, one member (i.e., the first author and interviewer) had significantly less work experience with people with complex support needs compared to all of the other team members, who had extensive professional experience in this field. This diversity in familiarity with the study population was valued by the research team as it hopefully reduced blind spots and preconceptions.

*2.3. Procedures*

Ethical approval was obtained from the Scientific and Ethical Review Committee of the Faculty of Behavioral and Movement Sciences, Vrije Universiteit Amsterdam, The Netherlands (registration number: VCWE-2018-003). The first author collected all of the data in a five-month period. All of the interviews were in-person and took place at locations preferred by the participant, which was at their home in all cases. The interviews were audio-recorded and lasted between 58 and 112 min, with an average duration of approximately 87 min. Each interview was transcribed verbatim and received a unique code. Confidentiality was assured by replacing all names with pseudonyms (i.e., R1, R2, R3, etc.) and by changing or removing all other identifying data (e.g., locale). The

transcripts remained in Dutch throughout the whole analysis and writing process. Only the quotations used in this article were translated into English. This was achieved through a back-translation procedure performed by one native English and one native Dutch speaker, both independent of this study, until high congruence between the original and back-translated quotes was achieved.

### 2.4. Interview and Pilot

The research team first constructed a semi-structured interview with broad and open-ended questions around the three BPNs: autonomy, competence, and relatedness. This interview was piloted with two fathers of young children with complex care needs. Feedback provided by the two fathers and their interview transcripts were discussed extensively within the research team, which eventually led to the formulation of several key questions for the final interview protocol. In this protocol, the participants were first asked to tell something about their relationship with their family member, for example how they communicated with each other, and how their family member enacted self-determined behavior. This was followed by three corresponding sets of questions, one for each BPN, which were administered sequentially. Each set addressed perceptions on the person-specific meanings of the BPN, the detection of (changes in) person-specific needs, recognition of (dis)satisfaction and motivation for person-specific needs, and the support and stimulation of person-specific needs (see Table 2 for the questions that guided each interview). Questions from the topic list were followed-up with prompts for elaboration and clarification. Because the abstract terms 'autonomy', 'competence', and 'relatedness' could be perceived as jargon by the participants, each question set started with an operational definition of the concept that was going to be discussed. Then, the participants were asked about the meaning of the need in their own lives, in order to facilitate the application of the concept to their family member with complex support needs. In order to encourage depth and detail, the interviewer provided extensive time for responses. In order to limit question-order bias, the interviewer alternated the order of the sets between interviews.

**Table 2.** Sample interview protocol.

| No. | Semi-Structured Interview Questions |
|---|---|
| 1. | What does autonomy [1] mean to you/What do you understand by it/How would you describe it? |
| 2. | What do you think autonomy means for (name family member)/What would (name family member) understand by this? |
| 3. | Can you tell me what changes there have been in the need for autonomy from birth to now, and how did you notice that? |
| 4. | How do you notice in (name family member) that he/she feels supported in his/her need for autonomy? |
| 5. | How do you notice in (name family member) that he/she does not feel supported in his/her need for autonomy? |
| 6. | How do you notice when (name family member) needs (more) autonomy? |
| 7. | What do you do to support or stimulate the need for autonomy? |
| 8. | What could healthcare professionals do to support or stimulate the need for autonomy/What possibilities do you see for this? |

[1] The same key questions were asked for competence and relatedness.

### 2.5. Data Analysis

Field notes taken during the interviews, as well as reflections and discussions of the data, were described in a diary as memos to support data analysis and code development [36]. All of the authors reflected on their own positionality and assumptions during all phases of the study with other team members and with outsiders. The interview transcripts were entered into NVivo 12 Pro software [38], and significant statements of the participants were coded line by line. The exploratory data analysis followed grounded theory strategies with the sensitizing concepts of autonomy, competence, and relatedness as the conceptual

framework [35,36]. This meant that although the three BPNs were the foundation for our conversations with the relatives, the coding was inductive and iterative. A constant comparative method was followed [39] in order to identify patterns and interrelationships in the perceptions of relatives, leading to the formation of the ultimate themes. As a result, these ultimate themes with the "thick description" of phenomena could include either one specific or multiple BPNs.

In the first phase of the data analysis, the first author assigned open codes to two interview transcripts by giving short descriptive terms to relevant statements that were used by the participants. The first author discussed these open codes with the third and second author individually, as well as with the entire research team together. These discussions led to the first clustering of codes in three categories: Interpretations of BPNs, Support options to find out BPNs, and Encountered obstacles. Then, the first and third author independently coded six interview transcripts, including the first two for a second time, through an iterative process of open and axial coding. Figure 1 illustrates this process of gradually creating related higher- and lower-level themes [40].

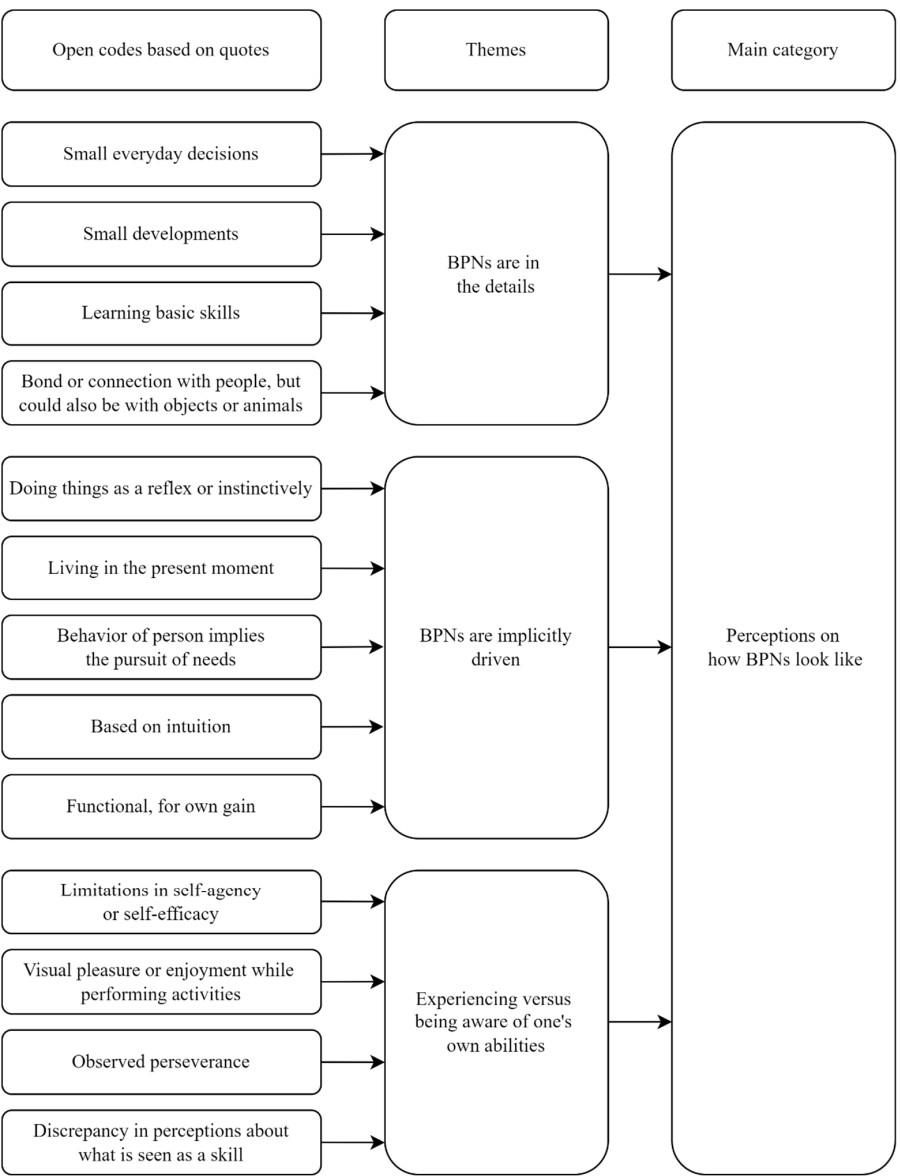

**Figure 1.** Open codes, themes, and categories reflecting the coding process of primary relatives' perceptions on what the BPNs of family members with complex support needs look like.

Throughout this analysis process, the coders compared, discussed, and refined all of the interim coding categories until they reached consensus before starting to code a new interview. They also actively searched for perceptions and statements that confirmed or contradicted previous findings (i.e., negative case analysis). Finally, the first author single-handedly coded the remaining interviews as described above, and again discussed and refined the findings during frequent meetings with the entire research team. The data analysis was completed when it was no longer possible to develop new themes or merge developed themes (i.e., when data saturation had occurred). At that moment, the final themes were structured under two main categories: (I) Primary relatives' perceptions on what the BPNs of family members with complex support needs look like, and (II) Primary relatives' perceptions on how support processes for the BPNs of family members with complex support needs work. In addition, the researchers incorporated perceptions on support processes into a tentative conceptual model to illustrate how relatives attempt to clarify and respond to their family members' specific BPNs.

## 3. Results

The purpose of this study was to understand primary relatives' perspectives on the meaning of satisfaction and motivation for the three BPNs (i.e., self-determination) for their family members with extensive support needs, as well as on finding ways to support them. The two overarching categories and corresponding themes that emerged through the data analysis are discussed below.

### 3.1. Primary Relatives' Perceptions on What the BPNs of Family Members with Complex Support Needs Look Like

The participants' descriptions of the extent to which their family members with complex support needs were able to direct or determine things for themselves varied from "he can determine very little . . . Actually, we (as parents) determine his life" (R1) to "she pretty much determines her daily schedule herself, insofar as that is possible" (R4). From their interpretations of the meaning of the BPNs, three themes were synthesized; they will be explained below.

### 3.1.1. BPNs Are All in the Details

Most of the participants stated regarding the nature of their family members' BPNs that needs were often very discrete, subtle, and idiosyncratic. Autonomy, for example, could be paraphrased as "the things" that the family members liked, they felt comfortable with, made them happy, they wanted, they did not want, they wanted differently, or they could choose for themselves. However, these specific things mainly revolved around small everyday activities. R3, for example, said "the bigger things in life, she cannot make decisions about those. But it is more about the small things in daily life . . . like when she does not want to get up, she will stay in bed." R7 mentioned: "well if he is on holiday for example, I always let him choose what to wear. Those are often little things, but yeah, I do notice that he has clear preferences."

Regarding relatedness, some participants indicated that "the others" with whom their family members felt connected, comfortable, or safe, and whom they liked, preferred, or interacted with, did not necessarily have to be a person but could also be a thing, object, or animal. R2, for example, stated about his son "I think that his feeling of connectedness is very broad, with everyone and everything, well, yeah, what moves . . . He is just really focused on other people and animals. Horses, dogs, it does not matter." Furthermore, this feeling of a connection with someone or something, or in a broader sense "belonging", was very noticeable, and could arise quickly in some family members, as in the aforementioned example of R2, while revealing itself less readily in others.

Competence was often explained as "the things" family members were able to do, were good at, were trying to do, or were learning to master. In this context, the participants spoke primarily about basic motor (e.g., holding an object or sitting up with(out) support),

communication (e.g., using pictograms), cognitive skills (e.g., knowing how to use a specific object), and social skills (e.g., waving to others). R8 mentioned, for example, "well then she helps me folding laundry, then I throw the washing from the dryer in the washing basket, I put it on the table and then she hands everything piece by piece." R7 explained this as "he really does have things in which he excels, but they are in general not skills that advance you in life or anything." Some participants even indicated that, due to the limited capacities of their family member, they would rather not speak about it to outsiders, like R1 who said:

> When people ask like "what can he do?" I say "he cannot do anything." Of course we know jolly well what he can do. But I do not need to say to a stranger, like "well he can stand in a standing-frame", because then they are like "right, he can take a few steps in a walking-frame, whatever", you know. I just say it very darkly like, "well he cannot do anything."

### 3.1.2. BPNs Are Implicitly Driven

The participants were often only able to infer the existence of specific BPNs rather than directly observing them. This meant that although certain behaviors implied that the family members were trying to express or pursue their BPNs, it often could not be determined why they had these specific needs, or in some cases even what exactly these specific needs entailed. For example, R2 described her son's drive to explore new skills (competence) as "he surprises us too sometimes, suddenly he can do something. He watches and copies something, and then he knows at once how to open something." According to the participants, BPNs were therefore mainly driven by instinct or primary drives, as R5 explained about her daughter's wishes (autonomy): "she lives in the present, so she, in that sense it is instinctive . . . So just about feeling NOW." R9 explained this as:

> What other people think and if they want something too . . . It just is not there. It is not that she says "that is not important to me", she just does not see it like that. It just does not exist for her. In fact a tremendously autonomous person. Yeah, autonomous to the core.

When R3 talked about her daughter's preferences for specific people (relatedness), she said: "that is very intuitive by (name of daughter), no signs are needed, no gifts are needed, she is very selective, purely intuitive." Another example was given by R9, who explained that her daughter "hardly shows interest at all in other people, other than functional, because she wants something . . . She will sit next to you because she likes to be cuddled, but it is never reciprocal."

On the other hand, some participants emphasized that specifically for the formation of a close bond (relatedness), the intensity and frequency of the caring relationship also played an important role. In other words, these participants indicated that they only noticed a relationship arising with people who were closely involved with their family members and had invested a lot of time to get to know them. R6 described this as follows:

> A relationship with his support workers, well there you see that it is kind of a family relationship. That he can just be himself (with them) and that they know him and that he has that attachment. That is, the more often he sees people, the more he can just be himself and find his own way in it.

### 3.1.3. Experiencing versus Being Aware of One's Own Abilities

The participants indicated that, in most cases, they doubted whether their family members were conscious of the personal capacities they had developed (competence). Some described this as an absence of experiencing the fact that one has the skills to accomplish a particular task (self-efficacy), as in the case of R9, who stated: "(I) wonder if she is aware of it (her abilities). It is also at a level of abstraction that I wonder, does she have that? And even if she has it, is it visible to us?" R7 mentioned that she could not notice any self-potential in her brother: "he does not expand it (his skills) like 'oh that is a strength of mine, I can develop that' . . . It is totally meaningless to him 'oh I am good at something.'"

Others even indicated that it often took a lot of effort to make family members aware that specific actions were self-generated (self-agency), and that these efforts were not always successful. R1, for example, described the importance of endless repetition in this process: "at school they know exactly what to do to stimulate him and they put him in it (a walking-frame) every day, countless times, and then suddenly the penny drops and he starts doing it, and then he likes it."

This observation of pleasure or gratification while performing an activity in itself without the presence of a deeper meaning or goal came up frequently during the interviews, and corresponds to what the German psychologist Karl Bühler [41] described as '*Funktionslust*'. R9, for example, said that her daughter clearly enjoyed riding a horse without experiencing it herself as something she can do: "the horse itself does not interest her either. Stroking or brushing or something like that, that is all boring. It is all about sitting on it."

On the other hand, some participants did report specific situations in which their family member with complex support needs appeared to be aware of personal capacities. As R4 described, "well that is very funny actually, because the moment that she realizes that it is very important that she shows how well she can communicate with (her) speech computer, she starts doing her utmost." Some participants, including the latter, even spoke explicitly of the presence of perseverance, which may indicate a desire of the family members to feel effective in what they do. R4 said, in this regard, that her daughter had a "very big drive to be able to keep walking", even when she had to relearn this after a scoliosis surgery "where she got metal rods in her body that reached deep into her pelvis", which prevented her pelvis from tilting.

Nevertheless, when the participants were able to detect a degree of self-efficacy, it often also revealed the discrepancy between the family members' perceptions about their own skills and how others interpreted these skills as such. R2, for example, explained that his son is quite successful and persistent at "opening things, climbing on things, or grabbing certain things he wants. So I think he feels, yeah, competent with that sort of things. However, it sounds crazy to me, as I would not think of it in that way myself." R5 illustrated this discrepancy as follows:

> *If you let (name of daughter) help you bake a cake or something . . . she is only able to put two or three things in a pan, but later she will tell someone else that she can bake, that she always does that, which is partly due to her getting a lot of positive feedback like "gosh how good of you (name of daughter)," you know? So (she thinks) "I can do that." But that does not touch reality.*

In addition, the participants indicated that in some cases this discrepancy could have negative or harmful consequences. R5, for example, said that even though she felt that her daughter deserved to experience the feeling of self-efficacy, "you also have to protect her when she expresses this (feeling) to other people, that they do not interpret it as such."

*3.2. Primary Relatives' Perceptions on How Support Processes for the BPNs of Family Members with Complex Support Needs Work*

The participants experienced the process of figuring out how their family member with complex support needs could be supported in BPNs as complicated and time-consuming. R6 described this as "you do not get a manual, you have to sense things, and keep an eye on how everything comes about and what affects what, the whole time." The participants' perceptions on how they attempted to support the BPNs were synthesized into three themes and a tentative conceptual framework, which will be explained below.

3.2.1. Detecting BPNs

BPN support started with noticing the person-specific expressions of their family members that represented their need for autonomy, competence, or relatedness. R2, for example, said "well yeah I think he can indicate all the things he needs for his goal in great detail, we just do not notice it all." The participants further indicated that BPN expressions or signals could vary according to context, were often idiosyncratic in nature, and could be

verbal or non-verbal (e.g., the presence or absence of certain sounds, behaviors, and other body language). In addition, expressions of BPNs were reactive in most situations. R1, for example, explained that on the rare occasions that her son expressed himself, he mainly did this by producing "higher sounds, louder sounds, faster sounds, (or) more sounds" in response to someone else's action or something happening at that moment that he liked or disliked (autonomy). R7 said about her brother:

> You have to look carefully at his behavior because, for example, if he is at his care-organization performing daytime activities, and he is bored there, then he will start to show difficult behavior, so to speak. Then he will no longer participate, or then he is more difficult for this support workers to handle. Yes, then you can see from his behavior that he is not feeling comfortable in his own skin and that he actually does not agree with what is happening.

On the other hand, a few participants indicated that signals could also be proactive. For example, the daughter of R9 took the initiative to express that she really wanted something (autonomy)—"because she does grab her coat herself and then she does put her coat on. Sometimes upside down and very often also the right way. And that is like 'well, shouldn't we head outside?'"—or that she liked specific social workers (relatedness) "who no longer came because they were going to do something else, that she then came to me with a picture 'where is she, he?' 'Sweet, sweet, sweet' (through using hand gestures)."

### 3.2.2. Clarifying BPNs

After signals of BPNs were noticed, the participants stressed the challenges and the importance of interpreting these signals correctly. This quandary is reflected in the statement by R4, who indicated that her daughter withdrew or fell asleep when she was dissatisfied, and that "this is very difficult in a group setting and difficult for therapists (to interpret) because they then say 'oh, she is tired', and I will say 'no, she is bored.'" R9 emphasized the importance as follows:

> If she cannot make clear what she wants, yeah nice idea autonomy, but I do not think you can do anything with it then . . . she can then only influence the things she can directly reach herself, she can grab, she can organize . . . Everything for which she dependents on other people, and that is a lot considering her developmental age, she needs communication for that.

In order to help clarify expressions regarding BPNs, the participants indicated that they often tried a wide variety of support resources and techniques, like using a "trial and error" method to rule different options out, calling in professional assistance, using supportive communication tools (e.g., speech computers, hand gestures, and pictograms), or relying on intuition and prior experiences with the family member. In this regard, R4 indicated that her understanding of her daughter's BPN expressions used to be very limited, and consisted only of relying on what her daughter was looking at. However, this improved drastically when her daughter learned how to use an eye-operated speech computer that made her go from "someone who cannot influence her own environment to someone who can." Nevertheless, some participants indicated that despite trying many different types of resources and techniques, signals often remained unclear to them, as in the situation illustrated by R1:

> We are working with pictograms, but that is still too challenging for him, but we do offer it to him. I notice myself in practice that I hardly do it, and because it just, yeah, is a fair amount of work and you do not get a response back. So that is pretty difficult . . . Signs, I did do them in the beginning . . . When he was only nine months old he got a hearing aid so then I had to use sign language too, but yeah, you can also do sign language to a cat, but they also will not understand it and (name of son) does not understand it either . . . With (name of son), the level is just too low to use signs. And he would never be able to make a sign back because he does not have the motor skills at all.

### 3.2.3. Creating Opportunities

All of the participants discussed family members' high degree of dependence on others to express, develop, maintain, and fulfil their BPNs. Because of this dependency, the participants stressed the importance of creating opportunities in all of these areas for their family member with complex support needs. This meant continuously creating optimal contexts, always putting the interest of family members first, and always striving for growth, development, and new opportunities. Regarding expressing one's own choices (autonomy), R9, for example, stated about her daughter: "(for me it is) extremely important, for the limited decisions she is able to make, that she is involved and has the right to say something about it." However, BPN expressions, as mentioned in the first theme, were usually reactive in nature, and therefore often only followed after another person's action or a change in the environment. Actively stimulating BPN expressions—for example, by offering options attuned to the family member (autonomy)—was therefore very important according to the participants. Similarly, the development of specific abilities (competence) was often only achieved by providing comprehensive learning opportunities over a long period of time, as R1 described:

> *I think he has had a walking-frame since he was three and he only now is starting to enjoy it and taking steps in it, but that is a matter of practice, practice, practice. Putting him in it every day at school . . . And well, yeah, he has his standing-frame, he stands in it every day at school, he practices with that, that is improving too, so yeah, he is developing in that. Just in teeny tiny steps.*

A caveat to this specific focus on learning opportunities was that such opportunities declined into adulthood, or as R3 explained about her 43-year-old daughter: "it is not so much that you are busy with the development. Actually, now I think about it I have not for a long time". R8 said about her sister "you know about that, she is done developing. She is 63 after all! Tried a lot and at a certain point, yeah, it is all done too . . . If it had been possible, it would have happened."

In the context of creating opportunities to fulfil clarified or well-known BPNs, such as facilitating what family members wanted (autonomy), the environment also played an important role. R4 shared, for example, that because of her daughter's well-known need to be around other people (relatedness), she deliberately took her to a reading hour for toddlers in the library and lets her visit a day-care facility for the elderly where she is seen as "the mascot." R1 explained the importance of her being the one who responds to her son's expressed need to be comforted (relatedness), relative to involved support workers:

> *If I sing a song when he cries, and I sing a certain song, then he is quiet, and if (name support worker) does that then he is not quiet. So that is well, then you just notice that he so to say, well, has a preference, does feel connected, right, to me, thankfully.*

The participants also elaborated on their challenges with creating opportunities and their inability to support expressed BPNs. Some participants indicated that because it was often impossible for them to notice or clarify current, new, or changes in BPNs, they often could only strive to make family members as comfortable as possible with their own life. R9 described this as follows:

> *I think the only thing we can go on is, is she acting happy, yes or no. And since she generally behaves as quite happy, shows happiness, we assume that apparently as far as she is concerned the world is okay. But what she would have wanted to do, other than what she is doing now, or how she would have wanted to live her life, I have no idea.*

Subsequently, when BPNs remained unclear, or were clear but unfeasible (e.g., because they were dangerous or inconvenient at the time), it was often difficult or even impossible for the participants to communicate this to their family member, or why this was the case, which could lead to dissatisfaction or frustration in both parties. R4 described this as follows:

> *There are also very difficult moments. For example, right before an operation and then say "I do not want to" (through using her speech computer). You know, she could not say that before. On one hand it is great, but now I have to do something with it.*

### 3.2.4. Tentative Conceptual Framework of BPN Support

In order to integrate the themes and codes from the data, a schematic flowchart was constructed (see Figure 2). This chart depicts the interactive steps and challenges of supporting the personal BPNs of people with complex care needs. The initial steps represent the idiosyncratic **proactive** or **reactive** expressions of a **current BPN**, respectively from the persons themselves or **induced by someone or something from the environment**. On the one hand, these expressions can **go unnoticed**, reducing the chance of fulfillment and inducing the chance of dissatisfaction or frustration. On the other hand, these expressions can **be noticed**, after which they can either be **interpreted correctly or be unclear**. When needs are unclear, it often prompts the **use of various resources or techniques** to assist clarification. When these **support resources fail repeatedly** for whatever reason, those involved can get stuck in a loop in which the BPN ultimately **remains unclear**, again reducing the chance of fulfillment and inducing the chance of dissatisfaction or frustration. When those involved can correctly interpret the BPN, it is either **possible or impossible to support it**, increasing the chance of satisfaction with the former and the chance of dissatisfaction or frustration with the latter.

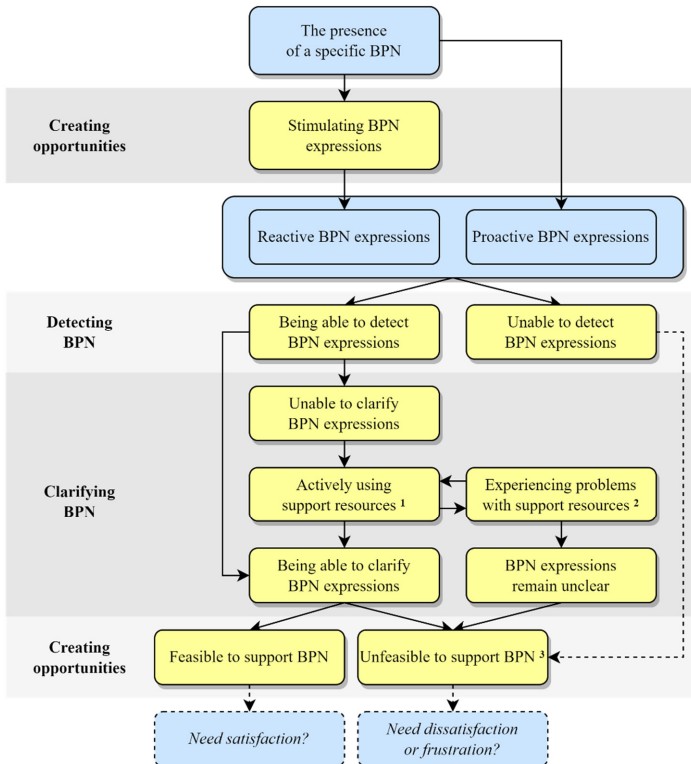

**Figure 2.** This figure represents a tentative theoretical flowchart of the steps and challenges of primary relatives in signaling and supporting the person-specific basic psychological needs of family members with complex support needs. The blue blocks represent the experiences and behaviors of the family members with complex support needs, while the yellow blocks represent the actions, experiences, and outcomes of the primary relatives. [1] Examples of types of support resources include therapy or training, time investment, trial and error, and communication tools (e.g., hand gestures, pictograms, and speech computers). [2] Examples of problems with support resources include the resources being too expensive, the resources having their own limitations, and regression when resources are not maintained. [3] Examples of the inability to create opportunities to support a need include the need being unclear, the need being dangerous, and the need not being executable.

## 4. Discussion

In line with Self-Determination Theory (SDT), this study explored primary relatives' perspectives on what they think satisfaction and motivation for the three BPNs mean to their family member with extensive support needs. First, all of the participants acknowledged the importance of the BPNs for self-determined (i.e., intrinsically motivated) action, as they described these as the things their family members liked, felt comfortable with, or wanted (autonomy); were able to do, trying to do, or learning to master (competence); and the others to which they felt attached, they made contact with, or they felt safe with (relatedness). Second, the relatives elaborated on the subtle and idiosyncratic nature of the BPNs in their family members; BPNs were often related to small everyday activities (autonomy); were mainly basic motor, communication, cognitive, or social skills (competence); and could arise with people as well as with things (relatedness). Third, the participants often had to infer a specific BPN from behavioral clues, instead of having it communicated directly. Fourth, some participants indicated that although they could not detect self-efficacy or even self-agency in their family members regarding their personal capacities (competence), they often did notice pleasure or enjoyment while performing these skills. Among the participants who could detect a degree of self-efficacy, there was often a discrepancy between how family members perceived their own skills and how the participants or others in the environment viewed these skills as such.

As seen from the perspective of Causal Agency Theory (CAT), people with complex support needs may thus sometimes develop "beliefs about the link between the self and the goal (control expectancy beliefs; 'When I want to do ___, I can')" [13] (p. 62). However, beliefs about one's own capabilities (i.e., capacity beliefs) and about factors that lead to goal attainment (i.e., causality beliefs), may be less apparent. In addition, people without disabilities, or with mild disabilities, are expected to strive towards their own needs (i.e., volitional action) and can ask for help when they experience problems doing so [13]. The self-determined behavior of people with complex support needs, on the other hand, may be more complicated because the actions to meet their own specific BPNs are much less visible to the outer world and therefore often go unnoticed, and are much more dependent on the actions performed and opportunities created by others. According to Wehmeyer [28], this increased reliance on others to perform certain behaviors is of minor importance in the ability to act volitionally. This dependency does, however, emphasize the importance of finding out what, when, and how people with extensive support needs perceive something as a desired outcome.

Based on the premise that self-determination develops and is maintained in interaction between the person and the environment [11], this study further identified how relatives' perspectives translated into support for the BPNs of their family member with complex support needs. The tentative model represented in Figure 2 describes how BPN-related signals, in interaction between relatives and their family members, in some occasions can lead to need fulfillment and thus self-determination. This complex process of relatives looking for, interpreting, and responding to BPN signals to the best of their abilities shows similarities with the concept of sensitive responsiveness from attachment theory, which indicates the extent to which a caregiver picks up on and adequately reacts to their infant's signals [42]. Specific for this population are the steps of the model in which relatives attempt to elicit expressions of BPNs and use diverse resources to clarify these expressions, and in which relatives need to deal with the fact that some of the BPNs will remain opaque to them. These steps will become easier over time for caregivers of typically developing children as their children develop increased capacities to properly identify and communicate their needs. However, they will remain challenging for the relatives of family members with complex support needs throughout the life course. Finally, these population-specific steps are in line with research by Vandesande et al. [7] on the preconditions and challenges that parents of children with severe or profound intellectual disabilities encountered when trying to establish a secure attachment relationship with their child. Examples of corresponding findings are (difficulties in) learning how to read non-verbal,

subtle, and idiosyncratic signals of their child; teaching them new things through patience and repetition; and using the help of professionals [7].

### 4.1. Implications for Practice

The environment of people with extensive support needs mainly consists of parents and healthcare professionals. Embregts et al. [43] showed that both parents and professionals are capable of taking the perspective of people with severe intellectual disabilities. However, De Geus-Neelen et al. [44] indicated that parents in this population evaluated their child's self-determination more positively than support workers did. De Geus-Neelen et al. [45] found significant discrepancies between relatives and support workers in their rating of the subjective wellbeing and internal needs of people with severe-to-profound intellectual disabilities. Furthermore, collaboration between relatives and support workers is hampered by the limited understanding of how the families of people with intellectual and developmental disabilities promote and support self-determination and its development in the home context [46].

The present study shows that BPNs, seen through the eyes of relatives, do not have a self-evident meaning for people with complex support needs, complicating support in realizing those needs. In order to support healthcare professionals in their knowledge about BPN satisfaction and motivation (i.e., self-determination), relatives may thus be involved more as equal partner in the client, caregiver, healthcare professional triad, as well as in education programs. In other words, healthcare professionals might accept the guidance of relatives and their family members in assigning meaning to their client's BPNs. Figure 2 can play a two-sided role in this. On the one hand, the figure may empower relatives to share what they know about how and what specific BPNs are communicated, how these expressions can be stimulated and recognized, what resources could be helpful for the clarification of expressions, and what optimal opportunities can be created to support specific BPNs. On the other hand, the figure can be used in conjunction with other tools, e.g., [47], to help relatives structure and articulate requests for assistance, such that professional support is more helpful. Furthermore, more structural awareness needs to be created among healthcare professionals that BPNs have different meanings compared to people without disabilities, and that support should always be in the best interest of the person with complex care needs, therefore requiring openness, sensitive responsiveness, dedication, and perseverance [48,49].

### 4.2. Limitations and Future Research

There are several limitations to this study. First, the task for relatives to discuss someone else's internal states is extremely challenging [50]. A study by Janssen et al. [51], for example, found that the parents of children with cerebral palsy rated their child's health-related quality of life, including autonomy, more negatively than the children did themselves. Nevertheless, the fact that the perspectives of relatives on meanings for their family members with complex support needs were colored by interpretations of the concepts for themselves possibly made the results more valuable, precisely because relatives are so important in helping their family members meet BPNs. Second, the interviews in this study were the sole data source, making validation through data-triangulation impossible. Future research might collect observations during situations related to self-determination, as well as member checks or focus groups with interviewed and non-interviewed relatives to verify the emerging themes and processes. Third, this study derived its sensitizing concepts from SDT. Although this study did not exclude relatives' perceptions that fitted better with other concepts, the use of other sensitizing concepts as a starting point, for example, related to one of the other mini-theories under SDT or to CAT, could have led to additional insights. Fourth, the tentative theoretical model is based on the processes of self-determination support from one specific person (i.e., a primary relative). However, if the family member with complex support needs lives in a household with several people, support could also be, for example, a joint process of relatives together. Fifth, although the

family members in this sample were relatively heterogeneous in their age and additional impairments (e.g., severity, type), their living situation was rather homogeneous, as the vast majority lived with the interviewed relative. It is possible that relatives like R7, whose family members mainly receive care from healthcare professionals, have different experiences and perceptions to relatives who are involved on a daily basis, like the others in this sample.

The purpose of this study was to investigate the perceptions of relatives, as they play an important role in the life of and have comprehensive knowledge of their relative with extensive and complex support needs [33,37]. However, relatives tend to differ from healthcare professionals in their views about people within this population [44,45]. Future research may therefore explore where these differences lie regarding the BPN satisfaction and motivation that go along with self-determination. Furthermore, inclusion was only based on the presence of a severe or profound intellectual disability. However, as people with complex support needs grow older, they build up life history and experiences. Accumulated experience, combined with the fact that relatives know their family members longer and thus probably better, could lead to changes or the crystallization of perceptions of the meanings of BPNs. Future research could look into how these changes develop over time (within people) and whether specific perceptions on meanings are tied to a specific age group or similar acquired experiences.

Additional findings showed that the three sensitizing concepts refer to needs that are highly intertwined. For example, learning to walk with a walking-frame (competence) can influence the family members' ability to determine whether, when, and where they want to go. Building a deep connection with someone (relatedness) can influence the family members' chance to be understood. An increase in competence or relatedness thus potentially facilitates autonomy satisfaction. Further research is needed into these associations between the three BPNs, in order to shed more light on how to create, increase, and extend a better supportive environment for people with complex support needs. Finally, the interviews also provided insights into a topic that went beyond the scope of this article, namely the specific meaning of autonomy, competence, and relatedness for relatives of family members with complex support needs themselves. Relatives, for example, indicated that they often felt limited or thwarted in their own BPNs due to the enormous burden of raising a family member with complex support needs. This aligns with other studies on caregiver burden for this specific group [5,6], and underlines the relevance of not only taking care of people with complex support needs but also their special caregivers.

## 5. Conclusions

This study showed that BPNs are recognized as relevant to self-determined (i.e., intrinsically motivated) action in people with complex support needs. However, the meaning of self-determination goes further than "making one's own choices." Enjoying and experiencing the things to one likes and meaningful interactions with others are equally important aspects. As explained by their relatives, BPNs in people with complex support needs are in the details, implicitly driven, and not always consciously or self-consciously expressed.

Although the environment is of great importance, supporting BPNs turns out to be complex. Limitations in communication and sensorimotor skills hinder the identification and clarification of needs, trapping those involved in a loop in which BPNs ultimately remain unclear, reducing the likelihood of fulfillment and increasing the likelihood of dissatisfaction or frustration. Due to the dependence on others, the environment also has a role in creating opportunities to fulfill needs. Relatives of people with complex support needs can help other caregivers who are learning to understand and support BPN signals, preferences and wishes.

The frequently encountered methodological difficulties in research on people within this target group [3] will continue to challenge the understanding of constructs within SDT for people with complex care needs and the support for these constructs by their

environment. Despite its limitations, however, the current study shows that relatives provide a unique insight into this quest. Based on the shared illustrative experiences of relatives and the resulting implications (e.g., involving relatives as equal partners, using the schematic flowchart, raising awareness that BPNs do not have self-evident meaning), we can now think more constructively about how to support people with complex support needs and their caregivers in promoting self-determination.

Taken together, in order for people with complex support needs to have equal opportunities to act volitionally and to give them "a stronger voice", it is of great importance that researchers, service providers, policymakers, and all other parties involved not only value and take good note of the intimate knowledge of relatives but also make good use of it.

**Author Contributions:** Conceptualization, J.M.v.T.v.S., A.M.W., P.J.C.M.E. and C.S.; methodology, J.M.v.T.v.S., A.M.W., A.d.l.C., P.J.C.M.E. and C.S.; validation, P.J.C.M.E. and C.S.; formal analysis, J.M.v.T.v.S., A.M.W. and A.d.l.C.; investigation, J.M.v.T.v.S.; resources, J.M.v.T.v.S.; data curation, J.M.v.T.v.S.; writing—original draft preparation, J.M.v.T.v.S., A.M.W. and A.d.l.C.; writing—review and editing, J.M.v.T.v.S., A.M.W., A.d.l.C., P.J.C.M.E. and C.S.; visualization, J.M.v.T.v.S.; supervision, A.M.W., P.J.C.M.E. and C.S.; project administration, J.M.v.T.v.S. and C.S.; funding acquisition, P.J.C.M.E. and C.S. All authors have read and agreed to the published version of the manuscript.

**Funding:** This research was funded by The Netherlands Organisation for Health Research and Development (ZonMw) Nationaal Programma Gewoon Bijzonder, grant number 845004005. Partial funding (i.e., co-financing) for this research was provided by Stichting Wetenschappelijk Onderzoek of 's Heeren Loo, The Netherlands.

**Institutional Review Board Statement:** The study was conducted in accordance with the Declaration of Helsinki, and approved by the Scientific and Ethical Review Committee of the Faculty of Behavioral and Movement Sciences, Vrije Universiteit Amsterdam, The Netherlands (registration number: VCWE-2018-003).

**Informed Consent Statement:** Informed consent was obtained from all subjects involved in the study.

**Data Availability Statement:** The data presented in this study are available on request from the corresponding author. The data are not publicly available due to ethical reasons.

**Acknowledgments:** The authors would like to express their sincere gratitude to the relatives and family members with complex support needs who participated in this study. Furthermore, recruitment was supported by 's Heeren Loo, EMB Nederland, and Per Saldo, three Dutch care and client advocacy organizations that support people with disabilities and their caregivers. The authors would like to thank these organizations sincerely for their time and effort in recruiting relatives.

**Conflicts of Interest:** The authors declare no conflict of interest. The funders had no role in the design of the study; in the collection, analyses, or interpretation of data; in the writing of the manuscript, or in the decision to publish the results.

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
