# Peer review of "Satisfying Basic Psychological Needs among People with Complex Support Needs: A Self-Determination Theory-Guided Analysis of Primary Relatives’ Perspectives"

_disabilities, doi:10.3390/disabilities2020024_

Round 1
Reviewer 1 Report
The issues discussed in the article have great cognitive and practical value. The Author/s focus on the psychosocial functioning of people with complex support needs. Research into this area is still insufficient, presumably due to methodological and recruitment difficulties. The image of the functioning of people with complex support needs, presented in the article, is indirect, as it was established on the basis of the experience of their non-disabled caregivers. Undoubtedly, multiple limitations and difficulties, including communication difficulties, may constitute a barrier to collecting material by means of direct verification of the experiences of this group of people with disabilities.
Remarks:
- I propose to organize the analysis of the collected material in the form of a table (tables) showing the main topics and specific categories, or possibly citations illustrating topics and categories.
- One respondent in the research does not live with their family but in a group home. This surely affects the image of the experiences of the person with a disability and their relatives.
- In my view, it is worth preparing an individual description of the respondents (practiced in qualitative research) ordering the cases (R1, R2). The possibility of referring to the context created by selected features, both of the respondent and the person with a disability, as well as the properties of their living environment, creates a greater opportunity to interpret certain experiences supported by quotes.
- Not all statements are included in the quotes.
- I suggest adding to the limitations of research the issue of significant age differences of people with disabilities. Age and related experiences gathered throughout life are important factors for the crystallization and disclosure of psychosocial needs and human competencies.
Author Response
Concerns: Revision manuscript disabilities-1659759
Date: April 11th, 2022
Dear Dr. Chen,
Thank you for the opportunity to revise and resubmit our paper, based on the constructive and helpful comments of the reviewers.
We have considered the reviewers’ comments and revised the manuscript accordingly. Changes include adjustments to the Table on demographics, an added Table with sample interview key questions and Figure with sample coding process flow, the split of long paragraphs (if possible), a relocation of the paragraph on research team expertise, more recognizable long quotes, additional limitations and suggestions for future research in the Discussion, inclusion of self-determination in the discussion of the results, and a stronger overall conclusion.
We believe that attending these matters has improved the manuscript. Please find our responses to the reviewer comments below and the tracked changes in the text of the manuscript.
We look forward to your response.
Kind Regards,
Jacqueline van Tuyll van Serooskerken, Agnes Willemen, Anne de la Croix, Petri Embregts, Carlo Schuengel
---------------------------------------------------------------------------------------
Reviewer 1
The issues discussed in the article have great cognitive and practical value. The Author/s focus on the psychosocial functioning of people with complex support needs. Research into this area is still insufficient, presumably due to methodological and recruitment difficulties. The image of the functioning of people with complex support needs, presented in the article, is indirect, as it was established on the basis of the experience of their non-disabled caregivers. Undoubtedly, multiple limitations and difficulties, including communication difficulties, may constitute a barrier to collecting material by means of direct verification of the experiences of this group of people with disabilities.
Remark 1
I propose to organize the analysis of the collected material in the form of a table (tables) showing the main topics and specific categories, or possibly citations illustrating topics and categories.
Response
We thank the reviewer for this valuable suggestion. We added a new Figure (Figure 1) to the data analysis paragraph in the Methods section (p. 8) that gives a visual representation of the coding process, illustrated for the first main category.
Remark 2
One respondent in the research does not live with their family but in a group home. This surely affects the image of the experiences of the person with a disability and their relatives.
Response
We thank the reviewer for pointing out this concern. We modified the original limitation in the Discussion about the almost homogeneous living situation in our sample and included this particular participant to address concern on possible differences in experiences and perceptions (p. 17, lines 637-680):
Fifth, although family members in this sample were relatively heterogeneous in their age and additional impairments (e.g., severity and type), their living situation was rather homogeneous (i.e., the vast majority lived with the interviewed relative). It is possible that relatives like R7, whose family members mainly receive care from healthcare professionals, have different experiences and perceptions than relatives who are involved on a daily basis, like the others in this sample.
Remark 3
In my view, it is worth preparing an individual description of the respondents (practiced in qualitative research) ordering the cases (R1, R2). The possibility of referring to the context created by selected features, both of the respondent and the person with a disability, as well as the properties of their living environment, creates a greater opportunity to interpret certain experiences supported by quotes.
Response
We thank the reviewer for this interesting suggestion. We adapted the table of demographic characteristics (pp. 4-5) from a combined description of the family members with complex support needs to an individual description, including additional respondent characteristics (i.e., relationship, primary caregiver of family member with complex support needs, and primary caregiver of other family members) to give more context to the quotes. In order to continue to guarantee the privacy of participants, we did chose to: 1. indicate the additional health problems on their basis of presence or absence instead of specifying them further per person, 2. omit the cause of the intellectual disability (previously indicated under ‘diagnosis’), as this is irrelevant to the context.
Remark 4
Not all statements are included in the quotes.
Response
We thank the reviewer for this comment, however, we are unsure whether the reviewer means that not all statements are enclosed in quotation marks or that not all interpretations are supplemented with quotes. With regard to the former, we made long quotes (those with more than 40 words) that do not need quotations marks according to APA more clearly recognizable, as can be seen on, for example, p. 9, lines 324-328. With regard to the latter, Lingard (2019), for example, stated that not all points in your argument need a quoted excerpt. We have therefore chosen to paraphrase or integrate a few statements of participants as part of the analysis, which makes the information in our opinion (more) manageable for the reader. Due to the measures that we took regarding validity, verifiability remains ensured.
Lingard, L. (2019). Beyond the default colon: Effective use of quotes in qualitative research. Perspectives on medical education, 8(6), 360-364.
Remark 5
I suggest adding to the limitations of research the issue of significant age differences of people with disabilities. Age and related experiences gathered throughout life are important factors for the crystallization and disclosure of psychosocial needs and human competencies.
Response
We thank the reviewer for this interesting suggestion. We added this at p. 16, lines 686-693:
Furthermore, inclusion was only based on the presence of a severe or profound intellectual disability. However, as people with complex support needs grow older, they build up life history and experiences. Accumulated experience, combined with the fact that relatives know their family members longer and thus probably better, could lead to changes or crystallization of perceptions on the meanings of BPNs. Future research could look into how these changes develop over time (within people) and whether specific perceptions on meanings are tied to a specific age group or similar acquired experiences (between people).
Reviewer 2 Report
Thank you for your submission, it was an interesting read and I enjoyed doing so. The needs of people with disabilities both psychological and physical are such a vital area for quality of life so i appreciate the conducting of this type of work. Please see my comments, feedback and suggestions below.
The introduction is very good, clear and well-referenced add well researched. The concepts covered have a key focus and the literature provided adequately foregrounds the wider project.
Some of the paragraphs are quite long and thus a little tough going for the reader ( see for example lines 69-97 or 231-255) - however this is more of a personal stylistic preference than an urgent change or anything like that so just something for you to consider but not something necessary prior to publication if the authors prefer the existing organsiation.
The ethics section is strong, clear and detailed - this is great and thus i have no ethical concerns about this study.
Interested that you have noted the research expertise of the team within your analysis - and your process of reflection /strengths consideration etc. Though this does attest to the clear expertise within the team and highlight important reflexivity, this may be better placed in your introduction or earlier in the methodology, perhaps in terms of the influences in your design procedure. This again is just a suggestion, a sit just feels a little out of place in a space for discussion of specifically your analytical process.
Figure 1 is good and useful as a visual representation, i would have expected to see it mentioned/introduced before its inclusion though. It does seem to be in the wrong place as it had split the sentence between lines 536 and 540 and i don't think this is intentional. This needs placing appropriately within the discussion surrounding it that seems to occur five lines or so later than its current position.
For such a detailed paper, the conclusions seem unfortunately brief in its current form - i suggest a restructure of your closing sections. Here i would have liked to see much more surrounding wrapping up and tying together the overall conclusions, reflecting on limitations and making recommendations for the future as you seem to have done in your implications for practice and your recommendations. I recommend that you either build a more detailed conclusions section out of the existing material that suits the wider paper, or integrate the conclusions in with your discussion at 4.1 and 4.2 to round off the paper. The content is largely present i just do think this could be better structurally.
Overall i recommend this paper for publication, i have ticked minor amendments to give you the opportunity to consider my suggestions but overall a very interesting and valuable paper that deserves to be published in my opinion. Well done.
Author Response
Concerns: Revision manuscript disabilities-1659759
Date: April 11th, 2022
Dear Dr. Chen,
Thank you for the opportunity to revise and resubmit our paper, based on the constructive and helpful comments of the reviewers.
We have considered the reviewers’ comments and revised the manuscript accordingly. Changes include adjustments to the Table on demographics, an added Table with sample interview key questions and Figure with sample coding process flow, the split of long paragraphs (if possible), a relocation of the paragraph on research team expertise, more recognizable long quotes, additional limitations and suggestions for future research in the Discussion, inclusion of self-determination in the discussion of the results, and a stronger overall conclusion.
We believe that attending these matters has improved the manuscript. Please find our responses to the reviewer comments below and the tracked changes in the text of the manuscript.
We look forward to your response.
Kind Regards,
Jacqueline van Tuyll van Serooskerken, Agnes Willemen, Anne de la Croix, Petri Embregts, Carlo Schuengel
---------------------------------------------------------------------------------------
Reviewer 2
Thank you for your submission, it was an interesting read and I enjoyed doing so. The needs of people with disabilities both psychological and physical are such a vital area for quality of life so i appreciate the conducting of this type of work. Please see my comments, feedback and suggestions below.
Comment 1
The introduction is very good, clear and well-referenced add well researched. The concepts covered have a key focus and the literature provided adequately foregrounds the wider project.
Response
We thank the reviewer for this compliment.
Comment 2
Some of the paragraphs are quite long and thus a little tough going for the reader (see for example lines 69-97 or 231-255) - however this is more of a personal stylistic preference than an urgent change or anything like that so just something for you to consider but not something necessary prior to publication if the authors prefer the existing organization.
Response
We agree with this valuable comment and have split a number of paragraphs into more concise arguments (e.g., p. 2, lines 85-86, p. 7, lines 268-269, p. 9, lines 303-304, 314-315). We believe that this his has improved the flow of the text for the reader.
Comment 3
The ethics section is strong, clear and detailed - this is great and thus i have no ethical concerns about this study.
Response
Again, we thank the reviewer for this compliment.
Comment 4
Interested that you have noted the research expertise of the team within your analysis - and your process of reflection /strengths consideration etc. Though this does attest to the clear expertise within the team and highlight important reflexivity, this may be better placed in your introduction or earlier in the methodology, perhaps in terms of the influences in your design procedure. This again is just a suggestion, a sit just feels a little out of place in a space for discussion of specifically your analytical process.
Response
We agree with the reviewer that the paragraph on reflexivity better suits earlier in the Methods section. The APA reporting standards for qualitative research (Levitt et al., 2018) advises to integrate the reflexivity paragraph in the Data sources and participants section. We followed this guideline and moved the paragraph to 2.2. (p. 4, lines 174-181).
Levitt, H. M., Bamberg, M., Creswell, J. W., Frost, D. M., Josselson, R., & Suárez-Orozco, C. (2018). Journal article reporting standards for qualitative primary, qualitative meta-analytic, and mixed methods research in psychology: The APA Publications and Communications Board task force report. American Psychologist, 73(1), 26-46.
Comment 5
Figure 1 is good and useful as a visual representation, i would have expected to see it mentioned/introduced before its inclusion though. It does seem to be in the wrong place as it had split the sentence between lines 536 and 540 and i don't think this is intentional. This needs placing appropriately within the discussion surrounding it that seems to occur five lines or so later than its current position.
Response
We thank the reviewer for this suggestion. The figure (now called Figure 2) is first mentioned in 3.2.4. (p. 13, line 550) and is now placed directly below this paragraph (p. 14).
Comment 6
For such a detailed paper, the conclusions seem unfortunately brief in its current form - i suggest a restructure of your closing sections. Here i would have liked to see much more surrounding wrapping up and tying together the overall conclusions, reflecting on limitations and making recommendations for the future as you seem to have done in your implications for practice and your recommendations. I recommend that you either build a more detailed conclusions section out of the existing material that suits the wider paper, or integrate the conclusions in with your discussion at 4.1 and 4.2 to round off the paper. The content is largely present i just do think this could be better structurally.
Response
We thank the reviewer for the invitation to add detailed information on the conclusions, limitations and recommendations. We have rewritten the Conclusions section as follows (pp. 17-18, 16, lines 712-740):
This study showed that BPNs are recognized as relevant to self-determined (i.e., intrinsically motivated) acting in people with complex support needs. However, the meaning of self-determination goes further than 'making one's own choices'. Enjoying and experiencing the things you like, and meaningful interactions with the environment are equally important aspects. As explained by their caregivers, BPNs in people with complex support needs are in the details, implicitly driven and not always consciously or self-consciously expressed.
Although the environment is of great importance, supporting BPNs turns out to be a complex process. Limitations in communication and sensorimotor skills hinder the identification and clarification of needs, trapping those involved in a loop where BPNs ultimately remain unclear, reducing the likelihood of fulfillment and increasing the likelihood of dissatisfaction or frustration. Due to the dependence on others, the environment also has a role in creating opportunities to fulfill needs. Relatives of people with complex support needs can help other caregivers learning to understand and support BPN signals, preferences and wishes.
The frequently encountered methodological difficulties in research on people within this target group [3] will continue to challenge the understanding of constructs within SDT for people with complex care needs and the support for these constructs by their environment. Despite its limitations, however, the current study however shows that relatives provide a unique insight into this quest. Based on the shared illustrative experiences of relatives and the resulting implications (e.g., involving relatives as equal partner, using the schematic flowchart, raising awareness that BPNs do not have self-evident meaning) we can now think more constructively about how to support people with complex support needs and their caregivers in promoting self-determination.
Taken together, for people with complex support needs to have equal opportunities to act volitional and to give them ‘a stronger voice’, it is of great importance that researchers, service providers, policymakers, and all other parties involved not only value and take good note of the intimate knowledge of relatives, but also make good use of it.
Comment 7
Overall i recommend this paper for publication, i have ticked minor amendments to give you the opportunity to consider my suggestions but overall a very interesting and valuable paper that deserves to be published in my opinion. Well done.
Response
We thank the reviewer for this compliment.
Reviewer 3 Report
Although the description of the data analysis is very detailed, and reflects the high reliability with which the information was systematically analyzed, a visual support could be used for a more synthetic explanation, for example, a diagram.
Can the questions that were asked in the semi-structured interview be presented?
The authors pertinently recognize the scope and limitations of the study, the highly accentuated inferential character of the interviews, neglecting the benefits of triangulating the information through other data collection strategies.
The issue of self-determination, addressed in the introduction, is not taken up again in the discussion of the results.
Author Response
Concerns: Revision manuscript disabilities-1659759
Date: April 11th, 2022
Dear Dr. Chen,
Thank you for the opportunity to revise and resubmit our paper, based on the constructive and helpful comments of the reviewers.
We have considered the reviewers’ comments and revised the manuscript accordingly. Changes include adjustments to the Table on demographics, an added Table with sample interview key questions and Figure with sample coding process flow, the split of long paragraphs (if possible), a relocation of the paragraph on research team expertise, more recognizable long quotes, additional limitations and suggestions for future research in the Discussion, inclusion of self-determination in the discussion of the results, and a stronger overall conclusion.
We believe that attending these matters has improved the manuscript. Please find our responses to the reviewer comments below and the tracked changes in the text of the manuscript.
We look forward to your response.
Kind Regards,
Jacqueline van Tuyll van Serooskerken, Agnes Willemen, Anne de la Croix, Petri Embregts, Carlo Schuengel
---------------------------------------------------------------------------------------
Reviewer 3
Comment 1
Although the description of the data analysis is very detailed, and reflects the high reliability with which the information was systematically analyzed, a visual support could be used for a more synthetic explanation, for example, a diagram.
Response
We thank the reviewer for this valuable suggestion. We added a new Figure (Figure 1) to the data analysis paragraph in the Methods section (p. 8) that gives a visual representation of the coding process, illustrated for the first main category.
Comment 2
Can the questions that were asked in the semi-structured interview be presented?
Response
We thank the reviewer for this interesting suggestion. We added a new Table (Table 2) to the Methods section (p. 6) that gives an overview of the topics that were mentioned in the semi-structured interview.
Comment 3
The authors pertinently recognize the scope and limitations of the study, the highly accentuated inferential character of the interviews, neglecting the benefits of triangulating the information through other data collection strategies.
Response
We thank the reviewer for this compliment.
Comment 4
The issue of self-determination, addressed in the introduction, is not taken up again in the discussion of the results.
Response
We thank the reviewer for the invitation to integrate the contextual theory of self-determination in the Discussion section. We did this at several places in the paragraphs, resulting in a better integration of the results within the theoretical model and relevant concepts (p. 15, line 568-575, 595-599, 603-608). We also integrated it more clearly in the Conclusion section (pp. 17-18, lines 712-740).